# Dynamics of Bacterial Diversity and Functions with Physicochemical Properties in Different Phases of Pig Manure Composting

**DOI:** 10.3390/biology12091197

**Published:** 2023-09-01

**Authors:** Xu Zhao, Juan Li, Hongxia Yuan, Zongxian Che, Lingui Xue

**Affiliations:** 1School of Chemical and Biological Engineering, Lanzhou Jiaotong University, Lanzhou 730070, China; zhaoxu6939438@163.com; 2Institute of Soil, Fertilizer and Water-Saving Agriculture, Gansu Academy of Agricultural Sciences, Lanzhou 730070, China; lijuan@gsagr.cn; 3Laboratory of Molecular Biology, Gansu Provincial Academic Institute for Medical Research, Lanzhou 730050, China; yuanhongxia520@126.com

**Keywords:** composting, pig manure, bacterial communities, diversity

## Abstract

**Simple Summary:**

China is not only the world’s largest country in terms of livestock and poultry farming, but also the number one country in terms of pig farming. Although the pig farming industry has made great achievements in China, it has also produced a large quantity of waste, generating about 2 billion tons of manure each year, accounting for 47% of total livestock manure. Environmental pollution problems are becoming increasingly prominent. Composting is an important, reasonable, and effective method to deal with manure, and it feeds nutrients and organic substances back into the soil, which can improve soil fertility and promote crop growth. Bacteria are key drivers in regulating ecosystem functions, and understanding the diversity and dynamic changes in bacteria in composting is very important for optimizing compost. This study investigated the bacterial community composition and functional change characteristics in the pig manure composting process and explored the correlation between the bacterial communities and physicochemical properties. These results will provide theoretical support for further optimizing pig manure compost conditions, promoting microbial metabolism, and improving the quality of organic fertilizers.

**Abstract:**

Bacteria are key drivers in regulating ecosystem functions, and understanding the diversity and dynamic changes in bacteria in composting is very important for optimizing compost. This study investigated the structure, composition, and function of bacterial communities in alkaline pig manure compost using Miseq sequencing, PICRUSt2. The ACE and Chao1 indices of the bacterial communities in various phases were significantly different. Bacterial communities of alkaline pig compost were different from neutral and acidic swine manure compost, and there were 438 genera of common bacteria in various stages. The main bacterium was the phylum *Firmicutes*. There were six genera, including *Romboutsia*, *Clostridium*, *Terrisporobacter*, *norank_f_Marinococcaceae*, *Saccharomonospora*, and *unclassified_f_Bacillaceae*, that were significantly correlated (*p* < 0.05), or even extremely significantly correlated (*p* < 0.001), with the physicochemical properties. TOC, moisture, C/N, and Tem were the key factors that caused changes in bacterial communities in composting. PICRUSt2 analysis showed that there were seven functional groups: metabolism (45.02–48.07%), environmental information processing (15.25–16.00%), genetic information processing (16.97–20.02%), cellular processes (3.63–4.37%), human diseases (0.71–0.82%), organismal systems (0.66–0.77%), and unclassified (13.93–14.36%). This study will provide a reference for improving bacteria growth and reproduction conditions in pig manure composting, optimizing the process, and improving the efficiency of composting.

## 1. Introduction

China is not only the world’s largest country in terms of livestock and poultry farming, but also the number one country for pig farming. In 2021, China had 491.93 million head of meat pigs in stock, accounting for more than half of the world’s capacity [1]. The breeding industry now accounts for more than half of China’s total agricultural output and has become a major source for the agricultural economy [2]. Although China’s pig farming has made great achievements, it has also produced a large amount of waste, generating about 2 billion tons of manure each year, accounting for 47% of total livestock manure. The environmental pollution problems caused by improper disposal of manure are becoming prominent [3]. Over 70% of pig feed is nitrogenous [4], and pigs do not fully digest these nitrogenous substances, resulting in a high nitrogen content in manure. If not composted and randomly piled, more than 80% of the nitrogen is discharged into the atmosphere in the form of ammonia, which not only pollutes the air but also leads to a reduction in the fertility of the compost [5].

Aerobic composting technology not only makes efficient use of the nutrients in livestock manure, but also uses high temperatures to kill worm eggs and most pathogenic bacteria to achieve the effect of deodorization and sterilization. Aerobic fermentation has the advantages of a large treatment capacity, low cost, simple operation, and easy promotion. The humus obtained from aerobic composting can be used in agricultural production, such as for forest fruits, vegetables, and organic planting, to increase the fertility of the land and reduce the amount of chemical fertilizers applied [6]. During composting, organic matter is decomposed by microorganisms. The metabolic activity of microorganisms is very active, and the heat generated can rapidly increase the temperature of the pile. The number of microorganisms and their structure and metabolic capacity vary from one raw material to another and from one composting period to another [7,8,9].

The composting process can be divided into four stages: The first stage is the initial stage, during which the temperature begins to rise and microorganisms begin to decompose organic matter. The second one is the thermophilic stage, where microorganisms metabolize fastest, organic matter is rapidly decomposed, and the temperature rises to more than 55 °C [10]. The third one is the bio-oxidative stage, where the temperature drops to around 40 °C. The fourth one is the maturation stage, during which the temperature drops below 40 °C. In this stage, the microorganisms continue to break down the organic matter and produce humus. The degree of compost maturity is determined by changes in temperature, pH, water content, total carbon/total nitrogen (C/N) ratio, organic matter content, and microorganism communities. Microorganisms play a dominant role in composting. It is important to analyze the changes in and diversity of microbial communities during composting in order to optimize the composting technology process [11].

In recent years, scholars have conducted in-depth studies on the structure and diversity of bacterial and fungal communities during composting [12,13,14]. But, metabolic function depends not only on the composition of the microbial community, but is also influenced by the environment in which they live [15]. It has been found that some bacteria are not only resistant to extreme environments (e.g., high pH and temperature) but also produce many extracellular enzymes that promote the mineralization of organic matter [16,17]. Some scholars have found that *Scytalidium*, *Myriococcum*, *Hyperthermophilic Pseudallescheria boydii*, and *Corynascus verrucosus* are present at different stages of composting and that *Aspergillus*, *Corynascus*, *Trichoderma*, *Penicillium Pseudallescheria* and *Batrachochytrium*, *Puccinia*, and *Rhizophagus* spp. are only present in the thermophilic stage of composting [18,19]. Mao et al. found that the dominant phylum, including *Firmicutes*, *Proteobacteria*, *Actinobacteria*, and *Acteroidetes*, existed in pig manure composting, but the relative abundances were distinct in different composting periods [20]. Neher et al. found that microorganisms showed different dominant species with different materials, methods, and composting stages [21]. Unfortunately, the biotransformation mechanisms of different materials are not yet known in composting, and the investigation of the mechanisms by which physicochemical properties affect the structure and function of bacterial communities will be a key step to investigate the biotransformation processes of composting.

With the rapid development of metagenomic and bioinformatics technologies in the past few years, microorganisms and their ecological functions in manure composting have been widely studied and reported. However, there are few reports on the mechanism of alkaline pig manure compost from the correlation between bacterial community changes and physicochemical factors and metabolic functions. In this study, composting was constructed using alkaline pig manure and wheat straw under natural conditions, aimed at determining the variation in bacterial communities in the composting process using high-throughput sequencing technology, and confirming the correlations between environmental factors and bacterial communities’ structures and abundances.

## 2. Materials and Methods

### 2.1. Composting and Sampling

Pig manure mixed with wheat straw at a ratio of 10:1 (weight basis) C/N ratio was adjusted to 22. Three composting piles were prepared in Wuwei, Northwest China in June 2021. The piles were approximately 15 m × 2.5 m × 1.5 m (length × width × height) and turned every 1 day in the first 15 days, then every 3 days until 35 days. The samples were collected at day 0, 3, 15, and 35, and then were sub-divided into three replicates. A portion of the sample was stored at −80 °C for the DNA extraction; the second part was stored at −4 °C to measure the ammonium, nitrate stored, moisture content, and seed germination indexes. The remaining sample was air-dried to measure the total nitrogen (TN) and total organic carbon (TOC). The composting process was divided into initial stage (T1), thermophilic stage (T2), biological oxidation stage (T3), and maturation stage (T4) at days 0, 3, 15, and 35, respectively.

### 2.2. Physicochemical Parameters Analysis

Temperature of composting was monitored daily. pH was determined using a pH meter after leaching using a fertilizer to water ratio of 1:5 (*w*/*w*). The moisture content, ammonium nitrogen, nitrate nitrogen, TN, TOC, C/N ratio, and germination index (GI) were determined using the methods described by Sun et al. and Duan et al. [22,23]. These physicochemical indicators were tested in triplicate.

### 2.3. DNA Extraction and 16S rRNA Amplification, Sequencing

Total DNA was extracted using a Power Soil DNA Isolation Kit (Mo Bio Laboratories, Solana Beach, CA, USA). Subsequently, the primer 338 F (5′-ACTCCTACGGGAGGCAGCAG-3′)/806R (5′-GGAC-TACHVGGGTWTCTAAT-3′) was used to amplify the bacterial 16S rRNA. PCR products were tested for the concentration, purity, and integrity of DNA using 2% agarose gel and NanoDrop 2000 spectrophotometer (Thermo Fisher Scientific, Waltham, MA, USA). And the Illumina MiSeq PE300 platform (China Shanghai Majorbio Biopharm) was used for high-throughput sequencing [24]. Bioinformatics analysis of sequencing results were used to obtain bacterial community composition and relative abundance. Sequences passing quality control were grouped into operational taxonomic units (OTUs) at 97% similarity level and sequencing result analyses were performed on the bioinformatics cloud platform (http://www.cloud.majorbio.com/ (accessed on 15 February 2023)) [25].

### 2.4. Statistical Data and Analysis

Shannon, Chao1, Simpson, and ACE were used to compare bacterial α-diversity, and PCA to compare β-diversity. Alpha diversity indices (Shannon, Chao1, and Observed ASVs) were obtained using Phyloseq package and contrasted using Welch’s *t*-test (*p* < 0.05). Cluster, Venn, and principal component analysis were conducted to compare the similarity of bacterial community composition. Principal component analysis (PCA) was performed to visualize the variation in community composition based on the Bray–Curtis distance and was conducted with Canoco 5.0 (Biometris, Wageningen, The Netherlands). Redundancy analysis (RDA) was conducted to reveal the relationship of multiple variations between environmental factors and community composition using Canoco for Windows software (v. 5.0). Spearman correlations with adjusted *p* value < 0.001 and absolute coefficient >0.8 within co-occurrence modules were visualized using the igraph package in R 3.5.1. The PICRUSt2 tool and genome (KEGG) database were used to predict bacterial metabolic functions [26]. All physicochemical properties and diversity indexes used for analyses were performed in triplicate. The data were analyzed using SPSS version 26.0. The significance level of differences was set at *p* < 0.05.

## 3. Results and Discussion

### 3.1. Changes in Physicochemical Properties in Composting

The physicochemical properties changed in the compost of pig manure are shown in Table 1. Temperature is one of the most important indicators of the intensity of the biochemical reaction in compost; the higher the temperature, the stronger the biochemical reaction and the faster the microbial metabolism. By detecting the temperature in compost, microorganism activity and decomposition degree can be determined [27]. The longer the composting is kept hot, the more harmless the pile, and the fewer pathogens there will be [28]. On the second day of composting, the temperature starts to rise rapidly, reaching 59.67 °C, with high temperatures (>50 °C) lasting for 17 days, reaching a maximum of 63.45 °C, and dropping to 38.53 °C on the 35th day. According to temperature, the composting process was divided into initial (days 0), thermophilic (days 1–3), bio-oxidative (days 4–21), and maturation (days 22–35) stages.

During the composting process, microorganisms used organic matter for reproductive growth and mineralization [29]. The TOC content decreased from 49.34 to 39.24%. In the T2 phase, the microorganism metabolic rate was the greatest, and the TOC content decreased the fastest. In the meantime, organic nitrogen decomposed, ammonification enhanced, and nitrification diminished. A large amount of ammonia was produced, resulting in lower TN and NO_3_^−^-N and higher NH_4_^+^-N. In the T2 phase, the NH_4_^+^-N content was the highest, and reached 3.71 mg·kg^−1^ TS. In the maturation phase (T4), the ammoniate weakened, nitrification enhanced, the TN and NO_3_^−^-N content increased, the TN content reached 2.94%, the NO_3_^−^-N content reached 1.503 mg·kg^−1^ TS, and the NH_4_^+^-N content reduced to 0.714 mg·kg^−1^ TS.

The moisture content (MC) decreased the fastest in the T2 and T3 phases, and and was slower in the T1 and T4 phases. At the end of composting, the moisture content decreased to 37.11%. The pH increased to 8.93 in the T2 phase, and decreased to 8.84 in the T4 phase. Since a large amount of NH_4_^+^ was produced in the T2 and T3 phases, in the T4 phase, microorganism metabolic activity decreased, ammoniate weakened, and the pH decreased and tended to be stable. C/N is one of the most important indicators for evaluating the maturity of compost [30]. In the composting process, the C/N ratio showed a first increased, then decreased tendency. In the T2 phase, the microorganisms consumed organic matter for reproduction and growth, and produced a large amount of NH_4_^+^ and NH_3_; thus, the TOC and TN content decreased rapidly, but the degradation rate of TN was higher than that of TOC, so C/N was raised. In the T4 phase, the TOC content decreased slowly, but the TN content increased, so C/N further decreased. At the end, C/N decreased to 15.39.

The GI value is an important indicator of the toxicity to evaluate the degree of decay in compost [31]. In the T1 and T2 phases, the seed germination was limited by high concentrations of NH_3_ and organic acids. Therefore, the seed germination index (GI) was 16.63% and 25.87% in the T1 and T2 phase, respectively. With the organic acids decreased and NH_3_ volatilized, the inhibitory activity was weakened, and microorganisms produced hormones that promote plant growth, such as auxin, gibberellin, etc., in the T4 phase [32]. At the end of composting, the GI increased to 64.48%. The higher the GI value, the lower the inhibition of seeds and the more effective the decay.

### 3.2. Changes in Bacterial Community Alpha Diversity in Composting

The Chao and Ace indices reflect the abundance of microorganism communities; the higher the value, the higher the abundance. The Shannon and Simpson indices describe the microorganism diversity; the higher the value, the higher the diversity. The Coverage index reflects whether the sequencing results represent the real situation of microorganisms in the samples [33]. The Coverage index (OTU coverage) of samples was greater than 99%, indicating that the sequencing results had high coverage and could truly reflect the microorganism community, and the results were credible. The Chao and Ace indices showed a gradually decreasing trend during the composting, with the highest at stage T1 and the lowest at stage T4, which indicated that the microorganism community abundance at stage T1 was higher than that at other stages. The Shannon index and Simpson index showed a trend of first increasing and then decreasing during the composting process, increasing at T2 and decreasing at T3 and T4. As the Shannon and Simpson indices also included the homogeneity of individual distribution among species in the calculation, the homogeneity of microorganisms in the samples increased during the high temperature stage, leading to an increase in the Shannon and Simpson indices, so the change trend was different from that of the Chao and ACE indices.

The common and unique bacterial genera at different stages of the composting process were evaluated using a Venn diagram, as shown in Figure 1. As time goes on, the number of bacterial genera gradually decreased, from 778 in the T1 stage to 612 in the T4 stage. The bacteria species at different stages of composting varied, but with 438 share genus, accounting for 49.83% of the total. There were 63, 16, 3, and 30 genera in stages T1, T2, T3, and T4, respectively, and represent 7.17%, 1.82%, 0.34%, and 3.4% of the respective totals. Different core microorganism communities play an important role at different stages of composting [34]. The alpha diversity of bacteria was highest in the initial phase; a lot of species did not take part in the fermentation and degradation process (Table 2).

### 3.3. Beta Diversity Analysis of Bacterial Communities

Beta diversity was used to assess the differences between microorganism communities. Principal component analysis (PCA) was used to compare the main factors of bacterial variation between treatment samples, with the more similar the sample composition, the closer the distance in the PCA plot [35]. The PCA analysis of the bacterial communities in samples at different phases is shown in Figure 2, with the contribution of PC1 and PC2 being 30.46% and 16.4%, respectively, with the sum of the contributions being 48.86%. Samples from T2 and T3 were close together, while T1 was further away from T2, T3, and T4, indicating that the bacterial communities were similar between the thermophilic and biological oxidation stages. The bacterial communities in T1 differed more from T2, T3, and T4. The T2 and T3 samples were closer together and had similar communities, while the T1 samples were further apart and, therefore, had more stable bacterial communities as the composting fermented. In addition, there were significant differences in temperature at different stages of composting, and the contribution of PC1 was significantly greater than that of PC2, indicating that temperature had a greater effect on the community composition of bacteria [36].

### 3.4. Changes in the Composition of Bacterial Communities

The distribution of bacterial communities at the phylum level in samples from different stages of composting is shown in Figure 3a. Succession of various microbial populations is a characteristic of the composting process, and nutrient access and oxygen availability are the major determinants of change in the microbe composition. Three major phyla, such as *Firmicutes*, *Actinobacteria*, and *Proteobacteria*, were the dominant groups in all samples, and the sum of their relative abundances accounted for more than 99%, with the highest dominance being Firmicutes. *Firmicutes* can form heat-resistant spores in high-temperature environments, and plays an important role in decomposing organic matter in the thermophilic stage of composting [37]. *Firmicutes*, *Actinobacteria*, and *Proteobacteria* accounted for 90.84%, 6.85%, and 1.10% of the total bacteria in T1. *Firmicutes*, *Actinobacteria*, and *Proteobacteria* accounted for 93.38%, 5.87%, and 0.54% in T2, respectively. *Firmicutes* and *Actinobacteria* accounted for 93.82% and 5.82% in T3, respectively. *Firmicutes*, *Actinobacteria*, and *Proteobacteria* accounted for 78.38%, 13.65%, and 6.63, in T4, respectively. *Firmicutes* slightly decreased in the T2 stage, while *Actinobacteria* and *Proteobacteria* increased in the T4 stage, and increased to 13.65%. *Proteobacteria* decreased in the T2 and T3 stages and increased to 6.63% in the T4 stage.

The microorganism abundance at the genus-level can reveal the succession of dominant microorganisms in composting [20]. The bacterial communities in samples from different phases of compost at the genus-level are shown in Figure 3b, with 23 bacterial genera, of which *Clostridium_sensu_stricto_1*, *unclassified_f_Bacillaceae*, *norank_f_Bacillaceae*, *Terrisporobacter*, *norank_f_Marinococcaceae*, *Pseudogracilibacillus*, *Corynebacterium*, *Turicibacter*, *Romboutsia*, *Saccharomonospora*, *Enteractinococcus*, *Oceanobacillus*, *Streptococcus*, *Sinibacillus*, and *Gracilibacillus* were the dominant genera. The bacterial community of pig manure changed significantly during the composting process, varying from one stage to another. The abundance of *unclassified_f_Bacillaceae* and *norank_f_Bacillaceae* were highest at T3 with 24.06% and 24.93%, respectively. The *norank_f_Marinococcaceae* genus was lowest at T2 with 0.62% and highest at T4 with 23.44%. It was found that *Bacteroides* and *Sporosarcina* were the main genera in the thermophilic stage (T2) of composting, but after the addition of bamboo biochar, the main genus changed to *Psychrobacter*, which shows that the physicochemical properties of materials have an impact on the microorganism community structure in pig manure composting [20,38].

### 3.5. Correlation between Bacterial Community and Physicochemical Properties

Composting is dominated by microorganisms, and the physicochemical properties of substrates have a strong influence on the structure of the microorganism communities [39]. The composition and distribution of microorganism communities are diverse in different phases of composting due to the physicochemical properties [40]. The correlation between the top 10 genera and physicochemical properties were evaluated using Spearman correlation coefficient. It is shown in Figure 4 that six genera, including *Romboutsia*, *Clostridium*, *Terrisporobacter*, *norank_f_Marinococcaceae*, *Saccharomonospora*, and *unclassified_f_Bacillaceae* were significantly correlated (*p* < 0.05), or even extremely significantly correlated (*p* < 0.001) with the physicochemical properties. *Clostridium_sensu_stricto_1* and *errisporobacter* were positively correlated with TOC and moisture, but *norank_f_Marinococcaceae* and *Saccharomonospora* were negatively correlated with TOC and moisture. *Saccharomonospora* was very significantly positively (*p* < 0.001) correlated with nitrate and GI; *norank_f_Marinococcaceae* was significantly positively (*p* < 0.05) correlated with nitrate and GI. *Clostridium_sensu_stricto_1* was very significantly negatively (*p* < 0.001) correlated with nitrate and GI; *Terrisporobacter* was significantly negatively (*p* < 0.05) correlated with nitrate and GI.

The correlation between physicochemical properties and bacterial genera on the relative abundance of the top 30 were evaluated at the genus level using RDA software (Canoco for Windows software (v. 5.0)), as shown in Figure 5. The length of the arrows in the RDA plot indicates the degree of correlation between the physicochemical properties and the distribution of the samples [41]. RDA1 and RDA2 together explained 90.98% of the total variation in bacterial community structure and physicochemical properties. TOC, moisture, C/N and Tem were the key factors that caused changes in bacterial communities in piles. Since C and N are the main nutrients for microorganism metabolism, C/N and TOC have a high positive correlation among the environmental factors that cause changes in bacterial communities [42]. Physicochemical property and bacterial community correlation was illustrated as follows: TOC and moisture > nitrate > GI > pH > temperature > C/N ≈ Amm-Nitrogen > TN. Therefore, the physicochemical properties could significantly influence the succession of bacterial communities in composting. Based on the bacterial community, moisture was the key factor in the T1 phase, while TN, temperature, and GI became the main influencing factors in the T2 and T3 phases. Generally, moisture was the most important environmental factor in the beginning of composting.

### 3.6. Bacteria’s Function in Composting

The reads obtained from the sequencing of compost samples were subjected to 16S rDNA function prediction, and we obtained the COG functional classification statistics plot shown in Figure 6a, where the horizontal coordinates are the different composting stage samples and the vertical coordinates are the ratio of the abundance of each function in each sample. As can be seen in Figure 6a, there were 24 functional groups in the samples at different stages. The changes in each functional gene were more stable during pig manure composting than replication, recombination and repair. The secondary metabolites biosynthesis, transport and catabolism, lipid transport and metabolism, carbohydrate transport and metabolism, transcription, general function prediction only, etc., were more significant.

The bacterial functions in composting were predicted by PICRUSt2 based on the KEGG pathway, as shown in Figure 6b. Most of the predicted pathways based on the relative abundance bacteria sequences in all samples could be divided into seven functional groups (pathway level 1): metabolism (45.02–48.07%), environmental information processing (15.25–16.0%), genetic information processing (16.97–20.02%), cellular processes (3.63–4.37%), human diseases (0.71–0.82%), organismal systems (0.66–0.77%), and unclassified (13.93–14.36%). Human diseases, metabolism, and organismal systems were highest in the T4 phase; cellular processes and environmental information processing were highest in the T3 phase; and genetic information processing was highest in the T3 phase. Chao Zhang et al. also obtained similar results in that green soybean hull composting was positively correlated with carbohydrate degradation genes [24].

The level 2 KEGG function predictions included 10 pathways for metabolism, 8 for environmental information processing, 5 for genetic information processing, and 4 for cellular processes among the pathways with a relative abundance over 1% (Figure 6c). The main metabolic pathways were carbohydrate metabolism (23.38–23.83%), energy metabolism (11.68–12.68%), and amino acid metabolism (24.02–24.6%). The cellular processes’ main pathways were cell motility (36.94–44.14%) and cellular processes and signaling (48.07–53.93%). The organismal systems, genetic information processing, and environmental information processing main pathways were enzyme families (59.2–63.52%), replication and repair (37.89–39.4%), and membrane transport (51.0–53.52%), respectively. As composting is a process in which microorganisms degrade organic matter such as proteins, amino acids, and sugars and release energy, the abundance of functional genes, such as carbohydrate metabolism, amino acid metabolism, and membrane transport, are relatively high in the different stages of composting [34].

## 4. Conclusions

The changes in bacterial communities and functions during alkaline pig manure composting were investigated in detail. The bacterial community structure and key groups differed in different stages, and bacterial groups were related to the physicochemical properties of compost materials. The changes in functions were related to bacterial communities’ succession. Metabolism, environmental information processing, and genetic information processing were the main pathways in composting. Our study will provide theoretical information for the optimization of pig manure composting technology.

## Figures and Tables

**Figure 1 biology-12-01197-f001:**
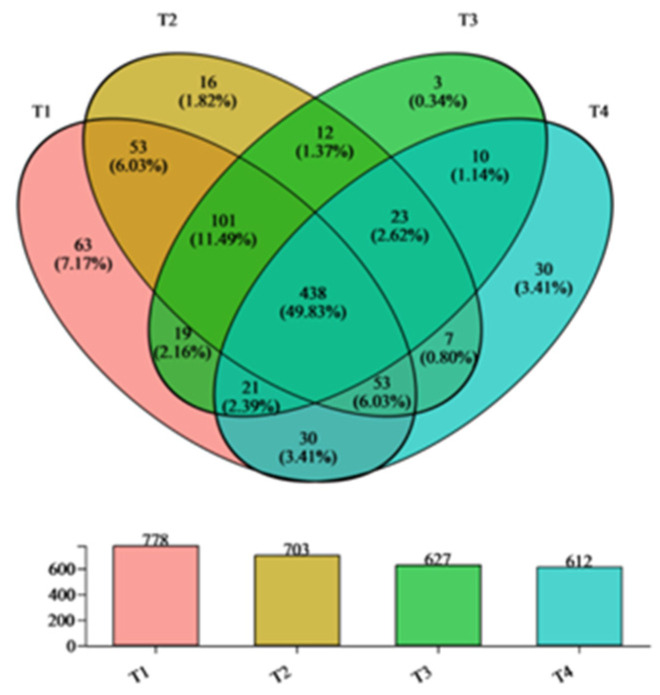
Venn diagram of bacterial communities.

**Figure 2 biology-12-01197-f002:**
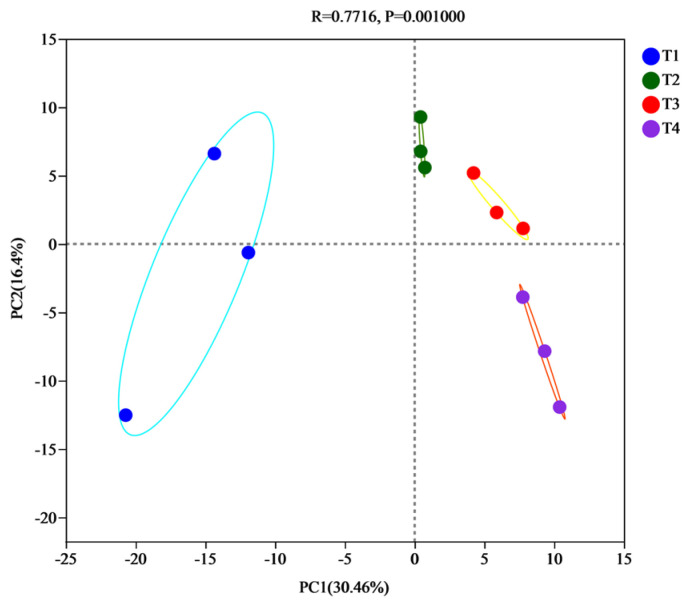
Principal component analysis (PCA). *X*-axis and *Y*-axis represent the two selected principal component axes, and the percentage indicates the value of the principal component in explaining the differences in the composition of the samples; the scales of *X*-axis and *Y*-axis are relative distances and have no practical significance; the dots of different colors or shapes represent the samples of different groupings, and the closer the dots of the two samples are to each other, the more similar is the composition of the species of the two samples.

**Figure 3 biology-12-01197-f003:**
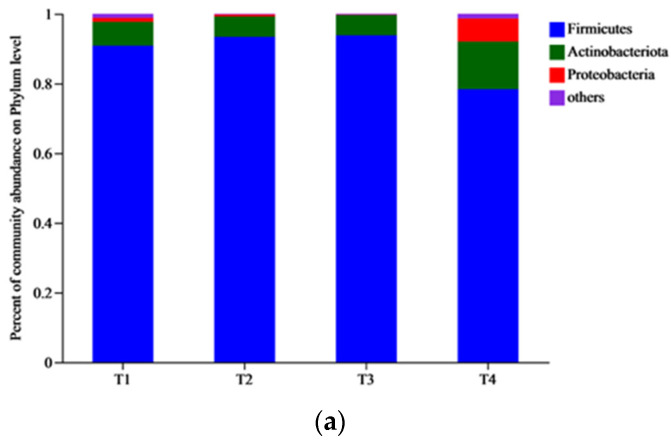
(**a**) The diagram of bacterial relative abundance at phylum level. Phyla with relative abundance <1% were combined together and indicated as “others”. Note: Horizontal/vertical coordinates are the sample name, vertical/horizontal coordinates are the proportion of the species in that sample, different colored bars represent different species, and the length of the bar represents the size of the proportion of the species. (**b**) The diagram of bacterial relative abundance at genus level. Genera with relative abundance <1% were combined together and indicated as “others”. Note: Horizontal/vertical coordinates are the sample name, vertical/horizontal coordinates are the proportion of the species in that sample, different colored bars represent different species, and the length of the bar represents the size of the proportion of the species.

**Figure 4 biology-12-01197-f004:**
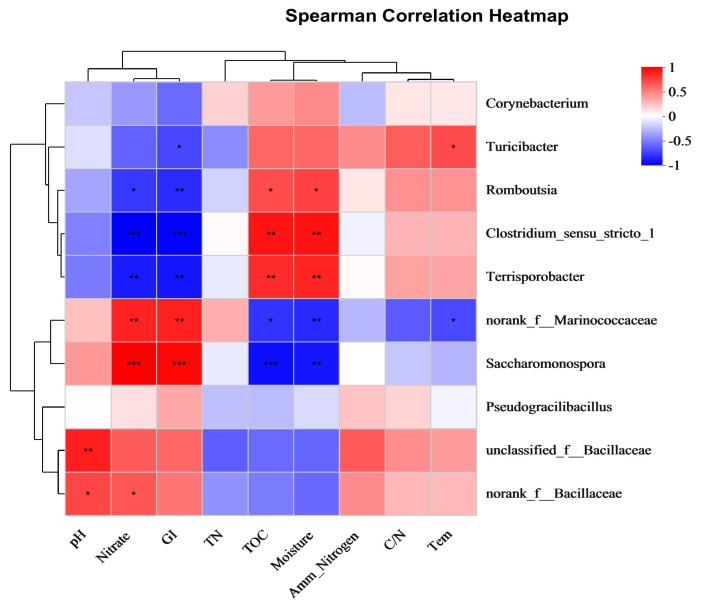
Spearman correlation between dominant genera and physicochemical metabolites during the composting process. The correlation coefficient is represented by the color and size of the circles. Dark red indicates positive correlation and dark blue indicates negative correlation. *p* values were calculated using Spearman’s rank correlation test, * *p* < 0.05; ** *p* < 0.01; *** *p* < 0.001.

**Figure 5 biology-12-01197-f005:**
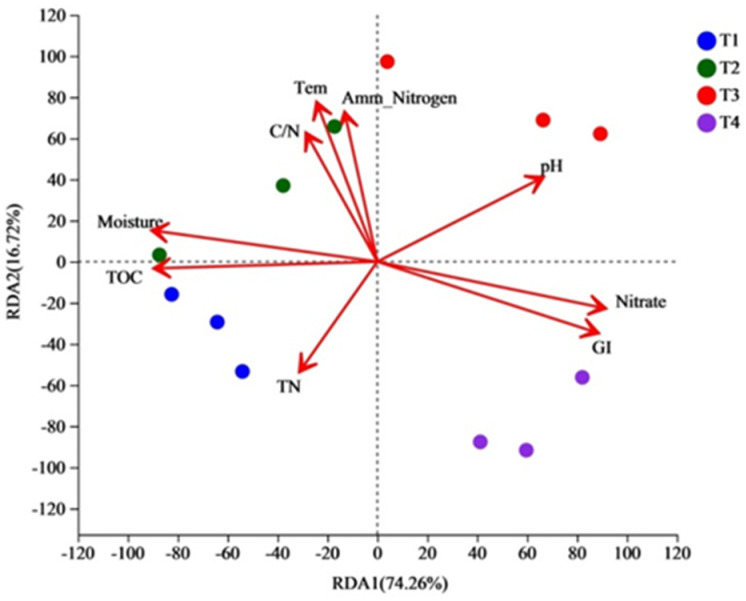
Multivariate redundancy analysis of bacterial and chemical properties (arrows) at different phases of composting.

**Figure 6 biology-12-01197-f006:**
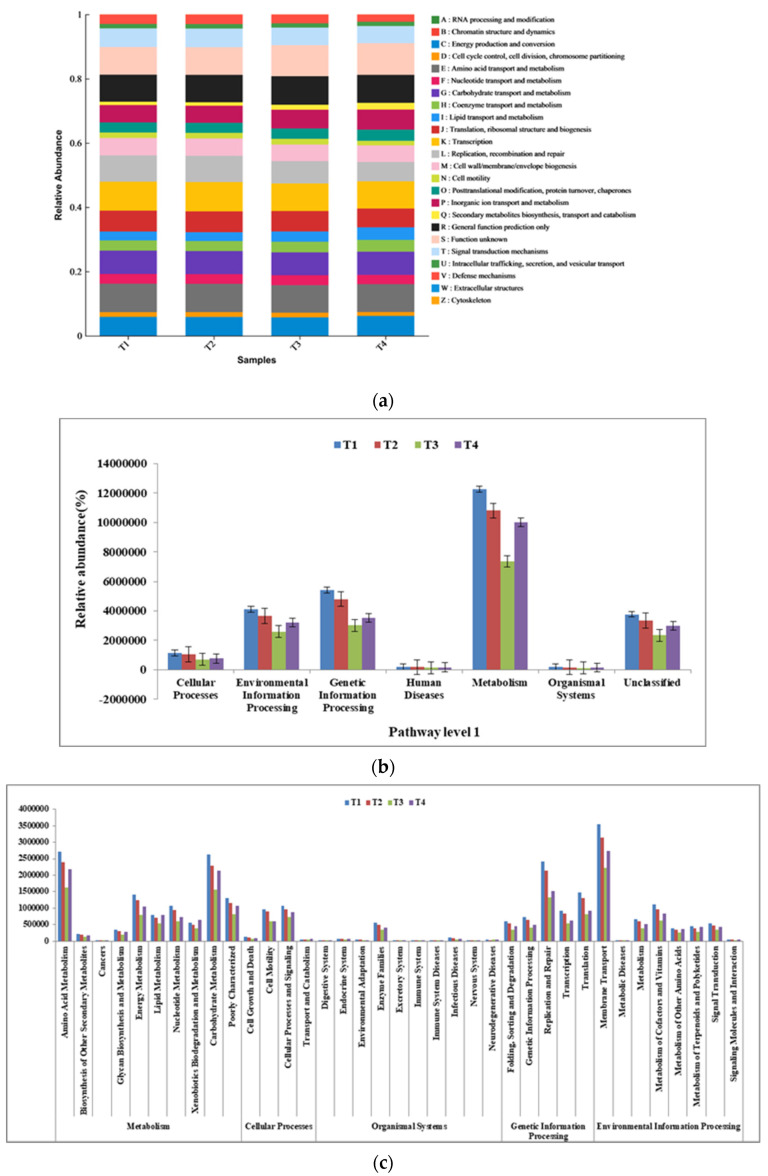
(**a**) Statistical diagram of COG functional classification. (**b**) KEGG metabolic pathway level 1 characteristics of bacteria. (**c**) KEGG metabolic pathway level 2 characteristics of bacteria.

**Table 1 biology-12-01197-t001:** Physicochemical properties of the compost in different phases.

Samples	Temperature (°C)	Moisture (%)	TOC (% TS)	TN (% TS)	NO_3_^−^-N (mg ·kg^−1^ TS)	NH_4_^+^-N (mg ·kg^−1^ TS)	C_N	pH	GI (%)
T1	39.43 ± 1.12 c	57.65 ± 1.83 a	49.34 ± 0.26 a	3.2 ± 0.103 a	0.058 ± 0.002 d	0.37 ± 0.082 d	15.34 ± 0.71 b	8.41 ± 0.04 d	16.63 ± 1.52 d
T2	59.67 ± 1.47 a	50.62 ± 1.76 b	47.44 ± 0.19 b	2.75 ± 0.78 d	0.398 ± 0.011 c	3.71 ± 0.146 a	17.26 ± 0.84 a	8.85 ± 0.07 b	25.87 ± 2.35 c
T3	59.63 ± 1.52 ab	43.19 ± 0.92 c	42.82 ± 0.36 c	2.84 ± 0.058 c	0.989 ± 0.026 b	2.19 ± 0.086 b	16.12 ± 0.47 c	8.93 ± 0.03 a	43.24 ± 2.88 b
T4	38.53 ± 1.18 d	37.11 ± 2.02 d	39.24 ± 0.43 cd	2.94 ± 0.063 b	1.503 ± 0.089 a	0.71 ± 0.033 c	15.39 ± 0.32 cd	8.84 ± 0.02 bc	64.48 ± 2.76 a

Note: Different lowercase letters in each column indicate significant differences in physical and chemical parameters in different periods of composting. Letters (a, b, c, d) indicate significant differences at *p* < 0.05 levels, respectively.

**Table 2 biology-12-01197-t002:** Alpha diversity of sample bacteria.

Samples	Coverage/%	Simpson	Shannon	ACE	Chao1
T1	99.776 ± 0.023 ab	0.118 ± 0.017 a	3.361 ± 0.145 b	699.583 ± 53.312 a	717.061 ± 39.267 a
T2	99.755 ± 0.035 cd	0.108 ± 0.031 ab	3.687 ± 0.212 a	697.129 ± 36.721 ab	650.47 ± 42.476 b
T3	99.758 ± 0.041 bc	0.090 ± 0.01 c	3.207 ± 0.201 c	639.142 ± 42.335 c	579.62 ± 25.261 c
T4	99.79 ± 0.016 a	0.065 ± 0.022 d	3.137 ± 0.163 cd	617.102 ± 78.635 cd	575.809 ± 56.012 cd

Note: Different lowercase letters in each column indicate significant differences between the same bacterial diversity indices. Letters (a, b, c, d) indicate significant differences at *p* < 0.05 levels, respectively.

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
