# Peer review of "Dynamics of Bacterial Diversity and Functions with Physicochemical Properties in Different Phases of Pig Manure Composting"

_biology, 2023, doi:10.3390/biology12091197_

Round 1

Reviewer 1 Report

Dear Authors, 

Many thanks for the great jobs. Please find attached the comments and suggestions for enhancing the manuscript.

Regards,

Dear Authors, 

Please find attached the comments and suggestions for enhancing the manuscript.

Regards,

Author Response

Dear Reviewers:

Thank you for your letter and for comments concerning our manuscript entitlee"Dynamics of bacterial diversity and functuions with physicochemical properities in different phases of pig manure composting"(ID:biology-2480710). These comments are all valuable and very helpful for revising and improving our paper,as well as the importent guiding significance to our researches. We have studied comments carefully and have made correction which we hope meet with approval. Revised portion are marked in red in the paper. 

Reviewer 2 Report

This study investigated the microbial community in various phases of pig manure composting. This can provide a reference for improving the quality of organic fertilizers. The manuscript was well organized and the data was sufficiently interpreted. However, the presentation should be improved before publication.

- Line 33: should be "various stages"

- Please refer to the following “Nomenclature of Microorganisms” to revise throughout the manuscript.

"Names of all bacterial taxa (kingdoms, phyla, classes, orders, families, genera, species, and subspecies) are printed in italics and should be italicized in the manuscript; strain designations and numbers are not."

https://jb.asm.org/content/nomenclature

- Line 33: should be "The main bacterial phyla were Firmicutes..."

- Fig 3: T1, T2, T3, and T4 should be explained in the figure legend.

- Fig 3 Y axis should be bacterial phyla relative abundance (%), it should be changed to 100% scale.

- The manuscript should be checked by a native English speaker.

- Bacteria names in figure should be italicized.

- Fig. 4: This is not bacterial genera. Please confirm.

English should be checked by a native speaker.

Author Response

Dear Reviewers:

Thank you for your letter and for the reviewer's comments concerning our manuscript. Those comments are all valuable and very helpful for revising and improving our paper. We have studied comments carefully and have correction which we hope meet with approval. Revised portion are marked in red in the paper.

Reviewer 3 Report

The manuscript offers an insight into the bacterial community during composting of pig manure. Why the authors decided to include ''Simple Summary'' section.  Family, genus, species should be italicized. The words ''bacterial'' and ''judged'' is used incorrectly in most cases in the text. Lines 38-42 are not part of an abstract. State separately the aim of the study. Please state the replicate samples used for the DNA extraction. Section 2.3 is incomplete. The germination index (GI) is not explained in the materials and methods section. Lines 162-163, 176-179 results are not presented in a scientific way. There is no discussion section. The analysis of the microbial community is inadequate. The references are incorrect. The text is hard to read and requires extensive English editing.

The overall quality of the manuscript is poor and it should be rejected.

Author Response

Dear Reviewer:

Thank you for your letter and for the reviewer's comments concerning our manuscript. Those comment are all valuable and very helpfull for revising and improving oue paper,as well as the important guiding significance to our researches.We have studied comments carefully and have made correction which we hope meet with approval.Revised portion are marked in red in the paper.

Round 2

Reviewer 2 Report

The manuscript can now be accepted for publication.

English is revised sufficiently

Author Response

Dear  Reviewer:

   Thank you for your letter and  comments concerning our manuscript entitled “Dynamics of bacterial diversity and functions with physico-chemical properties in different phases of pig manure composting” (ID: biology-2480710). Those comments are all valuable and very helpful for revising and improving our paper, as well as the important guiding significance to our researches. We have studied comments carefully and have made correction which we hope meet with approval. Revised portion are marked in blue in the paper. 

Reviewer 3 Report

The authors addressed most of the concerns

Please perform English editing of the manuscript

Author Response

Dear  Reviewer:

       Thank you for your comments concerning our manuscript entitled “Dynamics of bacterial diversity and functions with physico-chemical properties in different phases of pig manure composting” (ID: biology-2480710). Those comments are all valuable and very helpful for revising and improving our paper, as well as the important guiding significance to our researches. We have studied comments carefully and have made correction which we hope meet with approval. The English writing of the manuscript have checked and revised by my colleague. Revised portion are marked in blue in the paper. 
